

# Continuing challenges of elephant captivity: the captive environment, health issues, and welfare implications

Catherine Doyle[1], Heather Rally[2], Lester O'Brien[3], Mackenzie Tennison[4], Lori Marino[5,6] and Bob Jacobs[7]

[1] Performing Animal Welfare Society, San Andreas, California, United States
[2] Thrive Wild LLC, Ventura, California, United States
[3] Palladium Elephant Consulting Inc., Calgary, Alberta, Canada
[4] Department of Psychology, University of Washington, Seattle, Washington, United States
[5] Whale Sanctuary Project, Washington DC, United States
[6] Animal Studies, Department of Environmental Studies, New York University, New York, New York, United States
[7] Neuroscience Program, Department of Psychology, Colorado College, Colorado Springs, Colorado, United States

Corresponding author
Catherine Doyle,
cdoyle@pawsweb.org

## ABSTRACT

Although the well-being of elephants in captivity is of paramount importance, the confinement of these long-lived, highly intelligent, and socially complex animals continues to present significant challenges. Here, we provide an overview of the current state of elephant confinement (primarily in the West) by examining captive facilities, improvements, and continuing problems, and the clinical/behavioral/neural issues that remain. Specifically, we examine quantitative and qualitative aspects of the enclosed space, sociocognitive factors, dietary differences, and health/welfare concerns (*e.g.*, stereotypies, physical health, nutrition, reproduction, life expectancy). The challenges of the captive environment become especially salient when juxtaposed next to the complex, multifaceted characteristics of the elephant's natural environment. Despite the best efforts of some facilities to improve the captive environment, serious welfare challenges remain. Such confinement issues thus raise important welfare and ethical concerns with regards to captive elephant well-being.

## INTRODUCTION

Elephants, as "charismatic megafauna", have long been considered "problematic" candidates for captivity (*Hosey, Melfi & Ward, 2020*, p. 486), posing practical challenges and ethical concerns (*Clubb et al., 2008*; *Poole & Granli, 2009*; *Mason, 2010*; *Riddle & Stremme, 2011*; *Doyle, 2018*; *Rees, 2021*). Public attention continues to focus on their confinement and whether elephants' needs can be met to an acceptable level in captive situations, particularly zoos. Zoological professionals also are increasingly concerned about the effects of management practices on elephant well-being (*Greco et al., 2017*). Although many countries have laws that regulate the care of animals to better protect their welfare (*Rees, 2011*), the laws generally are not specific to individual species such as elephants.

Most elephant range countries (apart from India; *Central Zoo Authority (India), 2008*) do not have a national strategy addressing their captive elephant populations (*Riddle & Stremme, 2011*). With one exception (*e.g.*, see the UK Secretary of State's Standards of Modern Zoo Practice, which includes an appendix on the keeping of elephants in zoos— *Department for Environment & Food and Rural Affairs, 2012*), government licensing authorities do not generally have requirements for best practices specific to elephants. Some zoo accrediting organizations have developed elephant-specific requirements in order to standardize management and to establish a minimum level of care across zoos (*e.g.*, Association of Zoos and Aquariums, AZA; British and Irish Association of Zoos and Aquariums, BIAZA; European Association of Zoos and Aquariums, EAZA; Zoo and Aquarium Association-Australia, ZAA; Japan Association of Zoos and Aquariums-JAZA). Although such standards represent potential progress for the care of captive elephants, these facilities remain largely constrained by what is practicable in the zoo setting (*Clubb & Mason, 2002*), particularly in terms of space and social group size. Even under the highest standards, there remain unique if not formidable problems in meeting animals' essential needs in zoos (*e.g.*, space, social groupings, foraging opportunities, autonomy, choice, and cognitive challenges; *Morgan & Tromborg, 2007*; *Shepherdson & Carlstead, 2020*).

The scope of the problem for captive elephants is broad insofar as ~17,000 elephants are held in various types of captivity around the world (*Riddle & Stremme, 2011*; *Jackson et al., 2019*). About 14,000 of those are Asian elephants living outside of traditional zoos across 13 range countries, mostly in temples, logging camps, and tourism facilities (*Ghimire et al., 2024*). Globally, ~1,627 elephants are held in zoos. Zoos in Asia hold the highest number of elephants (~723), followed by Europe (~499), North America (~286), Latin America (~70), Australia (~26), and Africa (~25) (*The Elephant Database, 2024*). Of the ~10,000–12,000 zoos and animal parks in the world, fewer than 2.3% appear to be accredited (*Bacon, 2018*). Moreover, up to 95% of all zoos may not be meeting good practice standards for animal welfare (*Bacon, 2018*). Indeed, elephants are held in a variety of captive environments, typified by but not limited to zoos, as elephants are also kept in circuses, sanctuaries, and work/tourist camps. In the U.S., for example, there are ~365 elephants, with ~262 kept in zoos, ~47 in circuses or other commercial enterprises (*e.g.*, rides, encounters, private events), 21 in sanctuaries, and at least 35 in other facilities (*e.g.*, conservation centers) (*The Elephant Database, 2024*). Across captive facilities, there are significant differences in terms of space (both quantity and quality), substrate, amount of exercise, nutritional/husbandry needs, social/housing conditions, and level of human interactions (*Greco et al., 2016b*; *Shepherdson & Carlstead, 2020*). Although Western zoos have strived to address these conditions, including improving enclosures with more naturally appearing landscape immersion designs, environmental enrichment, and, in some zoos, more space and social opportunities, they are still wanting in many fundamental respects, especially for larger animals, of which elephants are the quintessential example (*Hancocks, 2002*).

Although the science of captive animal welfare is growing steadily, with increased attention paid to large mammals such as elephants (*Whitham & Wielebnowski, 2013*; *Shepherdson & Carlstead, 2020*), research indicates that some animals, including elephants,

do not fare well in captivity (*Clubb & Mason, 2003*; *Morgan & Tromborg, 2007*; *Mason, 2010*; *Fischer & Romero, 2018*). Although the terms "welfare" and "well-being" are sometimes used interchangeably, some authors make a distinction between the two (*Fraser, 1992*; *Martelli & Krishnasamy, 2023*). Animal well-being refers to the internal state of the individual whereas animal welfare encompasses management systems used to promote well-being (*Martelli & Krishnasamy, 2023*). In the present review, we are concerned not only with the effects of captive management systems (*i.e.*, welfare) but also the day-to-day physical, mental, and emotional health (*i.e.*, well-being) of the animals. There are both negative (*e.g.*, stereotypy, defined as invariant, repetitive movements with no apparent function; *Mason, 1991*; *Jacobs et al., 2021*) and positive welfare measures. As noted by *Miller et al. (2020)*, absence of negative welfare markers does not, in and of itself, indicate good welfare. For this reason, they suggest five positively-framed welfare markers, or "opportunities to thrive": (1) opportunity for a thoughtfully presented, well-balanced diet, (2) opportunity for self-maintenance (including shelter and species-specific substrates), (3) opportunity for optimal health (including a supportive environment that increases the likelihood of good health), (4) opportunity to express species-specific behavior (including quality spaces and appropriate social groupings), and (5) opportunity for choice and control to avoid suffering and distress and to make behavior meaningful (*Vicino & Miller, 2015*; *Miller et al., 2020*).

With these opportunities in mind, we reviewed the literature to examine the extent to which the care of elephants meets these opportunities to thrive, with the aim of identifying the challenges that remain in achieving them. This includes an overview of the behavioral and clinical factors for elephants in captivity. We focused on the most recent findings and representative facilities which, in many cases, are accredited by professional organizations (*e.g.*, AZA) and, as such, are considered to provide a higher level of care than other elephant-holding facilities (*Bansiddhi et al., 2020*). These accrediting organizations are found in the Global North (*e.g.*, Australia, Canada, Europe, Japan, United Kingdom, and United States) and their members hold ~56% of the elephants in zoos. However, we recognize that elephants are found in a variety of settings worldwide that present potential health and welfare problems, such as circuses and traveling shows, roadside attractions, rides, tourism, and controlled "encounters" during which people feed, touch or bathe elephants (*Kontogeorgopoulos, 2009*; *Grotto et al., 2020*; *Iossa, Soulsbury & Harris, 2009*). Although our focus is on accredited facilities, many of the issues we explore here (*e.g.*, stereotypy) are common to captive elephants around the world (*Vanitha, Thiyagesan & Baskaran, 2016*; *Bansiddhi, Brown & Thitaram, 2020*; *Bansiddhi et al., 2020*; *Fuktong et al., 2021*). Finally, we will suggest potential next steps for the well-being of elephants in captivity.

## LITERATURE REVIEW METHODOLOGY

We conducted a scoping review of the literature (*Munn et al., 2018*) on captive elephant welfare and on free-roaming Asian (*Elephas maximus*) and African (*Loxodonta africana*) elephants. Topics relevant to captive elephant welfare, both positive and negative, had been

previously identified in the existing literature: locomotion/walking rates/distances, body condition/health, stereotypies, sociality, play, foraging, rest, reproduction, housing, management (*e.g.*, enrichment, feeding, exercise), space and complexity, behavioral diversity, keeper-elephant relationships (*Asher, Williams & Yon, 2015*; *Meehan et al., 2016b*; *Chadwick et al., 2017*; *Williams et al., 2018*; *Meehan et al., 2019*; *Yon et al., 2019*; *Miller et al., 2020*; *Ghimire et al., 2024*). After a review of that literature, and with these factors in mind, we organized the article into physical and behavioral elements important to elephant welfare (*e.g.*, space and complexity, walking, sensory, physical environment, sociality, health issues, feeding, cognitive demands, human interactions), which became the basis for our searches. Subsequent searches were based on keywords that are characteristic of captive settings and important to the health and well-being of captive elephants: lifespan, mortality, welfare, sociality, enrichment, health, well-being, cognition, diet, nutrition, housing, zoos, exercise, digestion, reproduction, disease (*e.g.*, osteoarthritis, infectious diseases, obesity, foot disorders, dental issues), stereotypy, brain, sanctuaries, captivity, keeper-elephant relationships. We also incorporated information on free-roaming elephants such as habitats, ecology, sociality, foraging, diet, cognition, and behavior. Our primary searches were done through Google Scholar and PubMed. Other sources included our own publications and citation sections of published scientific papers. We selected only fully accessible publications. Exclusions included non-permanent facilities (*e.g.*, circuses), effects of elephants on zoo visitors, veterinary techniques, and commentaries. Considering new perspectives on welfare and the continuing challenges for elephants in captive situations, we excluded articles published prior to 2,000 except for those we considered foundational.

We used several sources of verifiable information in this review, with a total of 217 sources. The predominant source was peer-reviewed scientific literature (*n* = 165), followed by chapters in edited books (*n* = 24), scientific books (*n* = 3), and one newspaper article. Gray literature (*e.g.*, conference proceedings, white papers, government documents) was used sparingly (*n* = 14). Priority was given to the most recent findings so that the information in the article represents the most current overview possible. All but five citations were for publications after the year 2000, with 138 in the last 10 years (2014–2024), and 65 in the 10–20 year-old range (2004–2013). Webpages of professional organizations (*e.g.*, AZA) were accessed for information about their standards and guidelines. We focused on information about Western, mostly accredited, captive elephant facilities but did not exclude other information. Overall, our goal was to conduct a comprehensive search of factors relevant to the health and well-being of elephants in zoos, including findings that best represented current welfare issues for these animals. Moreover, we did not selectively exclude scientific findings that supported conflicting views about captive elephant welfare, as we wanted an accurate and complete picture of the status of captive elephants. To ensure we were conducting a comprehensive search, we examined the citation lists of all articles so that we did not omit any relevant article. Finally, when findings appeared in secondary sources (*e.g.*, review article, gray literature) we always confirmed them in the primary source.

## THE CAPTIVE ENVIRONMENT

The captive environment is multi-dimensional, encompassing several factors that affect both physical and behavioral elements of elephant welfare: captive space and complexity, movement, sensory experience and mental stimulation, diet and opportunities for foraging, and social opportunities.

### Enclosures

Although quantity of space is not explicitly mentioned in *Vicino & Miller*'s *(2015)* opportunities to thrive, it would seem to be a necessary component insofar as large, complex areas can sustain larger social groups of elephants, thus affording more opportunities for positive welfare by allowing choice, autonomy, and increased behavioral diversity. Alternatively, smaller elephant groups could also benefit from the positive welfare provided by larger spaces. Significant public attention has been directed at the housing and care of elephants (*Meehan et al., 2016b*), with that attention often focused on space. The scientific literature addressing free-living elephants indicates that, based on their large size and natural history, space is a key component of physical, behavioral, social, and mental well-being (*Poole & Granli, 2009*). *Meehan et al. (2016b)* note that space is a limited resource for zoos, and state that adequate space is required to support larger social groupings as well as robust feeding and enrichment programs; however, an "adequate" amount of space is not specified. An increase in space available to an elephant may also benefit metabolic health, but only if the increase in space is "substantial" (*Morfeld & Brown, 2017*, p. 11). Space limitations may be particularly problematic for male elephants, who range more widely in the wild than do family groups (*Hartley, Wood & Yon, 2019*). During musth, a period of heightened aggression and sexual drive (*Ghosal et al., 2013*), free-ranging males increase their range sizes and intermingle with multiple family units as they search for females in estrus (*Fernando et al., 2008*). Captive musth males also increase their movement and overall activity even though they remain constrained by their enclosures (*LaDue et al., 2022a*). Unlike free-ranging male elephants, captive males appear to have an elevated stress response during musth, although *LaDue et al. (2022a)* urge caution when interpreting wild-zoo differences due to sampling issues. Whereas providing access to females for mating may improve welfare, this may not be an option in all situations. It is worth noting that enclosure size is considered to be more important by scientists than by zoo personnel (*Gurusamy, Tribe & Phillips, 2023*). Just as the size of a species can affect its perception of enclosure complexity (*de Azevedo et al., 2023*), the size assessments that captive elephants make of their enclosures may potentially suffer from a floor effect insofar as all captive spaces may simply be too small from the perspective of elephants, whose home ranges are several orders of magnitude larger than even the largest zoo enclosure (*Atkinson & Lindsay, 2022*).

   In zoos, researchers have found that factors other than space have an effect on elephant welfare (*Meehan et al., 2016b*). In their epidemiological study of African and Asian elephant welfare in 68 accredited North American zoos, *Meehan et al. (2016b)* determined

that measures of space were not identified as risk factors for certain welfare outcomes (*e.g.*, stereotypies, obesity, female reproductive dysfunction). However, in accordance with a possible floor effect, they acknowledged that their investigation was "limited to the range of exhibit sizes at participating North American zoos [*i.e.*, 118 to 15,765 m$^2$], and future studies incorporating larger areas could potentially find associations between space and welfare outcomes" (p. 7). With relatively little variation in zoos, and with no zoos approximating even the smallest range space available to elephants in the wild, measuring the effect of zoo enclosure size on welfare is challenging (*Mason & Veasey, 2009*). In their natural habitat, elephants have expansive home ranges, extending from tens to 10,000 km$^2$ (*Fernando et al., 2008*; *Ngene et al., 2017*; *Bahar, Kasimi & Hambali, 2018*). Enclosure sizes for captive elephants vary. Some zoo accrediting agencies specify minimum space standards for outdoor spaces ranging from 500 m$^2$ per elephant (*AZA, 2021b*) to a minimum shared space of 3,000 m$^2$ (*BIAZA, 2019*). Minimum indoor stall size varies from 47 m$^2$ for one female elephant (*AZA, 2021b*) to 300 m$^2$ for four females (*BIAZA, 2019*), even though an elephant can spend a significant amount of time indoors (*Greco et al., 2016a*). Although some larger exhibits exist (*e.g.*, Disney's Animal Kingdom: 28,300 m$^2$; San Diego Zoo Safari Park: 13,000 m$^2$), the actual size of most outdoor enclosures in a sampling of 20 Western zoos (and three Eastern elephant centers) appears to be in the range of 17–6,937 m$^2$ per animal (*Taylor & Poole, 1998*), which is orders of magnitude smaller than their natural habitat—we note that more recent data on this issue are necessary. More recently, *Meehan et al. (2016a)* evaluated not only total exhibit size, but the individual elephant's experience based on the time they spent in different subdivisions of the exhibit. Calculated Total Space Experience (measurements of area as a function of time) values ranged from 117 to 15,742 m$^2$ (mean Overall Total Space Experience for African elephants was 3,642 and 1,781 m$^2$ for Asian elephants). For elephants in captive environments, given that an elephant can typically cover every section of a 10,000 m$^2$ enclosure in less than an hour, *Atkinson & Lindsay (2022)* suggest that an enclosure should be one or two orders of magnitude (*i.e.*, ~1 km$^2$) greater than traditional zoo exhibits. They note that such an enclosure then would allow for (1) a sufficient quantity and diversity of vegetation for normal foraging behaviors as well as visual screening, and (2) dozens of widely distributed focal points (*e.g.*, pools, rubbing rocks, sand mounds) for socialization and possible avoidance behaviors. Moreover, they suggest that a more tropical climate, similar to African and Asian elephants' natural habitat, would be preferable (as opposed to an arctic climate as in northern Canada). *Atkinson & Lindsay (2022)* conclude that range countries are the only places where elephants can truly flourish.

The issue of space in zoos is challenging, as zoos tend to be situated in urban centers, limiting possibilities for significant expansion. *Schmidt & Kappelhof (2019)* acknowledge that huge investments would be necessary to transform European elephant exhibits into complex fission-fusion housing systems and to create more capacity for male elephants. For zoos to maintain compatible herds, facilitate social behaviors, and manage male elephants, it would require "substantial financial investment, large physical spaces of many hectares [1 hectare = 10,000 m$^2$] and considerable expertise in elephant management" (*Hartley, Wood & Yon, 2019*, p. 74). Some zoos in the U.S. have

been investing in exhibit renovations at great cost, for relatively small expansions. The Fort Worth Zoo recently invested $32 million to improve their 12,141 m² elephant enclosure (*Fort Worth Zoo, 2024*); the Cincinnati Zoo will spend $50 million for a 20,234 m² area (*Miller, 2023*); and the Memphis Zoo recently announced a $250 million plan to renovate their exhibits for African elephants, rhinoceroses, and giraffes (*Moore, 2024*). Future research will be required to determine if these renovations have the effect of significantly improving elephant well-being. For now, improvement in zoo association elephant standards for space, social groupings, and male elephants may force zoos to invest in exhibit changes. In addition, research conducted in non-zoo facilities that provide much larger, more complex spaces (*e.g.*, sanctuaries, conservation centers, semi-captive situations) could potentially improve our understanding of the effect of these spaces on welfare, with applications to zoos and other captive situations.

## Space complexity and enrichment

The attributes of an enclosure would seem to be an essential element in *Vicino & Miller*'s *(2015)* requirement that animals have quality spaces to express species-specific behavior. In this regard, space complexity has indeed been shown to be important to the welfare of elephants in zoos. *Scott & LaDue (2019)* examined the behavior of two elephants in three different treatments: a small simple space (with a pool and shade structures), a small complex space, and a large complex space (the latter two with a pool, shade structures, elevated feeders, and varied substrate). Results indicated an increase in foraging and self-maintenance and a decrease in time spent stationary in both the small and large complex spaces, underscoring the importance of space complexity. *Scott & LaDue (2019)* determined that increased exhibit complexity was sufficient to significantly improve behavioral diversity, overall activity levels, and other behaviors indicative of positive welfare, leading them to conclude that complexity should be prioritized over space (see also *DiVincenti, McDowell & Herrelko, 2023*). However, the authors' own results show that activity and behavioral diversity were highest in the large complex space. Increased investigatory behavior was significantly higher in only the large complex yard, with stereotypic behavior decreasing "drastically" in that condition. This indicates that the combination of space *and* complexity achieved even more impressive results than did complexity alone. *Atkinson & Lindsay (2022)* suggest that more expansive spaces, with varied topography, natural substrates, and natural vegetation for foraging, can better provide the level of environmental complexity necessary to stimulate natural behaviors and provide choice, autonomy, and diversity of experience. A similar conclusion was reached by *Brown et al. (2020)*, who noted better elephant welfare in enriched, less constricted, and more stimulating environments.

Creating space complexity can include the addition of various forms of enrichment to an enclosure in order to promote the expression of important behaviors (*e.g.*, foraging, self-maintenance) and provide opportunities to engage in exploration, play, and problem solving, as well as exercise choice (*Hoy, Murray & Tribe, 2010*; *French, Mancini & Sharp, 2018*; *Greco et al., 2016b*). Importantly, in their natural habitat, animals use their cognitive skills to overcome problems that may directly or indirectly affect their survival.

Unfortunately, this type of problem solving has historically been absent in captivity (*Meehan & Mench, 2007*). Only in more recent years has greater focus been placed on more cognitive types of enrichment, including time-delay puzzle feeders, high tech enrichment devices, and insightful problem solving tasks (*Foerder et al., 2011*; *Highfill et al., 2016*; *Krebs & Watters, 2017*; *French, Mancini & Sharp, 2018*; *Barrett & Benson-Amran, 2020*). Such cognitive enrichment, especially when species-appropriate abilities are stimulated, appears to provide some positive behavioral benefits by increasing the animals' interest in their environment (*Krebs & Watters, 2017*; *Holland, 2018*). Nevertheless, the long-term benefits of such enrichment remain unclear. Furthermore, cognitive enrichment may be provided infrequently in zoos, with keepers perceiving that semi-permanent and permanent exhibit features (*e.g.*, pools, logs, scratching areas, puzzle feeders) are more important than cognitive enrichment (*Hoy, Murray & Tribe, 2010*; *Greco et al., 2016b*). *Greco et al. (2016b)* found that problem-solving opportunities were components in the enrichment programs of 97% of the 63 zoos they examined; however, these opportunities were provided infrequently. In fact, the frequency of problem-solving enrichment was so low that *Greco et al. (2016b)* were unable to test its effects in their welfare models. Such a low frequency may be attributed to the time required to prepare and distribute enrichment, as opposed to feeding enrichment, tactile enrichment, and human-animal interactions (*Hoy, Murray & Tribe, 2010*).

There are few studies on the effects of enrichment for elephants, with most of them focused on feeding enrichment (*Greco et al., 2016b*). As such, more research is necessary in this area. Just as importantly, zoos must focus on the suite of enrichment options available to them, especially in the area of problem-solving enrichment. Impediments to the provision of enrichment include conflicting priorities, lack of available time, and uncertainty about which enrichment practices are most effective (*Hoy, Murray & Tribe, 2010*; *Tuite et al., 2022*). This indicates the need for more support at all staff levels and for increased resources to allow for the effective implementation and evaluation of enrichment (*Hoy, Murray & Tribe, 2010*).

## Exercise

*Vicino & Miller*'s *(2015)* third opportunity to thrive is that the animal should have an environment that promotes good health. Certainly, an important welfare issue for elephants is the amount of exercise they can experience, which is often measured by distance traveled (*Holdgate et al., 2016*). Estimates of walking distances in nature vary by age and sex (*Slotow & van Dyk, 2004*), season (*Loarie, van Aarde & Pimm, 2009*), and resource availability (*Gadd, 2002*). Nevertheless, ~8–12 km/day are normal, with much greater distances (up to ~50 km/day) being common (*Wall et al., 2013*; *Miller, Hogan & Meehan, 2016*). Zoo-based studies of walking in elephants do not use uniform outcome measures, making comparisons to natural habitat studies difficult. At the San Diego Zoo Safari Park, *Miller, Andrews & Anderson (2012)* and *Miller, Hogan & Meehan (2016)* estimated an average walking distance of 8.65–9.82 km/day for African elephants. Similarly, *Brady, McMahon & Naulty (2021)* estimated Asian elephants traveled 9.35 km/day at the Dublin Zoo (enclosure size: 8,500 m$^2$). *Rowell (2014)* calculated a distance of

9.05 km over an 18-hour period at the Royal Melbourne Zoological Gardens (enclosure size: 5,143 m$^2$). *Holdgate et al. (2016)*, who gathered data on both African and Asian elephants across 30 North American zoos, concluded that elephants walked only a distance of 5.3 km/day, far shorter distances than in the wild. In general, the larger the enclosure, the greater the opportunity for purposeful walking. Importantly, *Holdgate et al. (2016)* also state that the range of exhibit sizes in the study population may not have been sufficient to show the effect that very large areas can have on walking distances, and that elephants might walk more if provided larger enclosures than are currently available in North American zoos (cf., *Meehan et al., 2016b*). *Holdgate et al. (2016)* further determined that, overall, the distances elephants walked in zoos were influenced most significantly by feed-related factors. However, *Hacker, Miller & Schulte (2018)* note that *Holdgate et al. (2016)* compared walking across institutions and not within the manipulated space of one facility. In their own study, *Hacker, Miller & Schulte (2018)* manipulated space and food in different conditions: access to two yards with food (Both); access to one yard with food (Half); access to both yards with food in only one (Both/Half). They then measured walking distances and behavior among elephants belonging to different dominance groups. Increased walking distances for middle-ranked African elephants and a significant correlation between the change in behavioral diversity across the Both and Half treatments, along with dominance ranking, led the authors to conclude that both food and space impact elephant walking and behavior at their institution. More clarity is needed to assess the importance of space relative to food and how it affects walking distances.

Although the functional need for walking appears to be reduced in zoos due to the provision of social and nutritional resources, ~60 million years of evolution has provided Proboscidea with a range of anatomical and physiological specializations for long distance living (*Shoshani, 1998*; *Poole & Granli, 2009*). For example, elephant foot morphology allows them to load the most lateral aspect of their feet when walking which, along with decreasing pressure below the foot's fat pad, helps them maintain foot health (*Panagiotopoulou et al., 2016*). This locomotion mechanism does not work well for elephants housed in small enclosures with hard substrates (*Panagiotopoulou et al., 2016*). The exercise elephants obtain by walking is also a crucial component of an enriched environment because it supplies the brain with oxygenated blood, increases the effectiveness of the immune system, and enhances an animal's cognitive abilities (*Jacobs et al., 2021*), while also providing metabolic benefits (*Morfeld & Brown, 2017*). Finally, *Holdgate et al. (2016)* contend that walking is important because it supports exploratory behavior and its information-gathering function may be rewarding outside of the acquisition of resources.

Clearly, more research is required to determine the relationships between walking, space, food, and social conditions for elephants in zoos. However, the lack of significantly larger enclosure sizes will continue to constrain such investigations. As previously stated, research in non-zoo facilities with larger, more complex spaces could enhance our knowledge, potentially improving captive elephant welfare or helping to determine whether elephants should be kept in captivity at all.

## Sensory experience

Estimates of elephants' walking distances in zoos often miss a critical element: how the captive elephant's sensory experiences while walking differ from those of a free-ranging elephant whose sensory environment is composed of a broad and often-changing variety of sounds, smells, sights, and tactile and taste sensations. In nature, elephants exist in different types of large, dynamic ecosystems (*e.g.*, forests, open savanna, wet marsh, and desert; *Clubb & Mason, 2002*), where they interact with diverse ecological features and collect information about the physical and social environment. In captivity, sensory experiences are largely predicted by the temporal and spatial features of their enclosures, which are relatively limited and unchanging, and the enrichment provided within these limited spaces (*Lucas & Stanyon, 2017*). Elephants in zoos typically are confined to one or more outdoor yards, smaller holding areas, and/or a barn with individual and/or group stalls (*Lucas & Stanyon, 2017*). Outdoor enclosures may contain a variety of natural and artificial features such as soil and hard substrates (*e.g.*, concrete), water features, wet and dry wallows, and (concrete) rockwork or tree features to accommodate maintenance behaviors (*Clubb & Mason, 2002*; *BIAZA, 2019*; *AZA, 2021b*). Note that concrete substrate/ rock/tree features challenge *Vicino & Miller*'s *(2015)* requirement that the opportunity to self-maintain includes substrate that is species-specific. If sensory stimulation *via* exploratory behavior and information-gathering in the captive environment is indeed rewarding in itself (*Holdgate et al., 2016*), more diverse and natural surroundings with the opportunity for more varied sensory experiences may be important to improving elephant welfare. Facilities providing this type of environment currently include three Global Federation of Animal Sanctuaries accredited elephant sanctuaries (two in the U.S. and one in Brazil) and the AZA accredited White Oak Conservation Foundation in Florida. For example, the elephant area at White Oak measures 68.8 km$^2$, encompassing forest, open grasslands, ponds, and wetlands, and a variety of plant (food) species (*White Oak Conservation, 2022*). The Elephant Sanctuary in Tennessee spans 12.4 km$^2$ and features spring-fed lakes, pastures and woodlands, and natural forage (*The Elephant Sanctuary in Tennessee, 2024*).

Insofar as human visitors are another prominent feature of the zoo environment, they are also part of an elephant's sensory experience. Decades of research indicate that the effect of zoo visitors can be negative, neutral, or positive in terms of animal behavior and welfare (*Hosey & Melfi, 2014*; *Sherwen & Hemsworth, 2019*). Species-specific differences, the nature of the visitor interactions, enclosure design, and individual animal temperament may contribute to the variation in zoo animal responses (*Sherwen & Hemsworth, 2019*). Visitor effects on elephants are also mixed, as are the facilities and conditions under which they were studied. Based on research at the St. Louis Zoo, *Krishnan & Braude (2014)* concluded that large audiences could serve as a source of behavioral enrichment, as indicated by decreased stereotypies and the elephants' use of space. However, their definition of a "high" crowd size was five or more people and "low" as fewer than five, raising the question of whether their methodology was sensitive enough to support their conclusion. *Manning et al. (2023)* studied elephants in a South African tourism park that

offered human-elephant interactions (*e.g.*, walks, feeding); observations were conducted before, during, and after COVID pandemic closures. They concluded that changes in behavior, including elevated self-directed behaviors (a novel indicator of anxiety in African elephants), demonstrated that returning tourist numbers, whether high or low, were stressful for the elephants. In examining public feeding of elephants in a zoo, *Fernandez, Upchurch & Hawkes (2021)* suggested that public feedings may function as enrichment for some elephants. Although the study was limited, the authors found that public feedings appeared to result in increased social activity and a decrease in stereotypies (only one elephant engaged in the behavior regularly) when compared with nonpublic feeding days for two of three elephants. All three elephants showed increased foraging and were more active in the period after a public feeding session. *Williams et al. (2023)* found that elephants had positive responses to visitors more frequently than would be expected by chance; however, they noted that enclosure design may reduce the negative impact of zoo visitors on elephants because of the necessity of keeping a safe distance between visitors and elephants. Although *Quadros et al. (2014)* did not specify elephants in their study of the effects of visitor noise on 12 mammal species, they found significant behavioral differences at the individual level. They concluded that the noise generated by zoo visitors negatively affects the welfare of individual animals, especially when noise amplitude exceeded 70 dB(A). Given the popularity of elephants in zoos, and the noise that can be generated by the public (especially in indoor spaces), further study is clearly required to determine whether and in what ways such human interaction affects elephant welfare across different exhibit designs.

Caretakers represent another human facet of an elephant's environment. Because elephants are highly managed, keepers are essential contributors to elephant welfare (*Carlstead, Paris & Brown, 2019*), an important aspect of *Vicino & Miller*'s *(2015)* opportunity for optimal health. Elephants typically spend just over 50 percent of their daytime hours under behavioral control in managed activities (*e.g.*, exercise sessions, foot and skin care, training; *Greco et al., 2016a*). There appears to be an inverse relationship between time spent with keepers and an elephant's rate of stereotypy, even outside of time spent in keeper-directed activities. *Greco et al. (2016a)* attribute this to the social relationships that elephants form with their keepers. Keepers may also benefit from these relationships, as the stronger the bond felt by keepers, the less likely they were to report dissatisfaction with their jobs (*Carlstead, Paris & Brown, 2019*). However, keepers generally appear to develop stronger bonds with Asian rather than African elephants, which is reflected in job satisfaction measures (*Carlstead, Paris & Brown, 2019*). Management and housing differences may account for such measures (*Carlstead, Paris & Brown, 2019*) because Asian elephants are managed for a significantly greater proportion of their day than African elephants, who tend to be more independent (*Greco et al., 2016b*). On average, African elephants spend more time in unique environments, and experience greater total space and more outdoor space than Asian elephants (*Meehan et al., 2016a*). Bonds between keeper and elephant may be related to welfare benefits for both. *Carlstead, Paris & Brown (2019)* found that positive keeper attitudes were associated with lower mean serum cortisol concentrations in elephants, suggesting that good keeper-elephant

relationships lower stress responsiveness in elephants. In the future, zoos could substantially expand exhibits for Asian and African elephants, potentially giving the animals greater autonomy and requiring less keeper-managed time. This may call for reassessing the nature of keeper-elephant relationships in order to maintain a beneficial relationship for both keepers and elephants. Another important area of reassessment has been the end of bullhook use in AZA zoos (*AZA, 2022*) and the phase-out of these devices in EAZA facilities (*EAZA, 2019*), as their use presents serious challenges to the safety and well-being of both elephants and keepers (*Wilson et al., 2015*).

Because the study of keeper-elephant relationships has received little attention in zoos (*Carlstead, Paris & Brown, 2019*), more research in this area is necessary. Research that focuses on the attitudes and beliefs of keepers relative to welfare measures in elephants could contribute to positive welfare for elephants and ensure that keeper-elephant relationships are mutually beneficial (*Carlstead, Paris & Brown, 2019*).

## Sociality

Facilitating appropriate social structures for captive elephants is a particular challenge for *Vicino & Miller*'s *(2015)* provision that the opportunity to express species-specific behavior requires appropriate social groupings. Free-ranging elephants tend to live in matriarchal, multi-generational family groups of two to 10 adult females and their offspring (*Sukumar, 2006*; *Vance, Archie & Moss, 2009*; *de Silva, Schmid & Wittemyer, 2017*). Elephant family groups share a fission-fusion structure, separating and merging with larger groups of up to several hundred elephants (*Poole & Moss, 2008*; *de Silva, Ranjeewa & Kryazhimskiy, 2011*). In nature, female elephants remain with their natal herd, forming strong lifelong bonds with related females, although family fissions are also common (*Garai, 1992*; *Sukumar, 2006*; *Poole & Moss, 2008*). Males differ in that they remain with their family group until sexual maturity, when they disperse (*Lee et al., 2011*). They nevertheless maintain complex social ties with conspecifics of all ages and both genders (*Hartley, Wood & Yon, 2019*).

Historically, captive elephants have often been limited to small groups of mostly unrelated adult females with very few infants or juveniles (*Clubb & Mason, 2002*; *Rees, 2009*), making the inability to replicate natural social structures an area of concern for the welfare of elephants in zoos (*Williams et al., 2019b*). Current research suggests that captive elephants should be held in related, multi-generational groups for optimal welfare (*Williams et al., 2019b*; *Finch et al., 2020*; *Harvey et al., 2018*; *Hartley & Stanley, 2016*). In North America, only 20% of the 226 adult elephants in U.S. zoos studied by *Meehan et al. (2016a)* had the opportunity to interact with young elephants, albeit an absence of calves in a group does not necessarily lead to poor welfare (*Williams et al., 2019b*). Currently, zoo standards allow for as few as three or four elephants (*AZA, 2021b*; *BIAZA, 2019*) and some zoos continue to house females singly (*AZA, 2021a*). The AZA's three-elephant minimum does not specify that all elephants must be females, which can contribute to isolating female elephants (*e.g.*, two males housed separately from one female), or even the same species. *Schmidt & Kappelhof (2019)* report that most of the female reproductive elephants in the Asian Elephant European Association of Zoos and Aquaria Ex situ Programme (EEP) are being held in family groups, an improvement over previous years. However, in

general, there has been uneven progress in increasing mean group size for both Asian and African elephants across North American and European zoos, with significant increases found only for African elephants in North America (*Rees, 2021*). In North American zoos, mean group sizes remain at 4.45 and 3.3 for African and Asian elephants, respectively (*Rees, 2021*). Although *Williams et al. (2019b)* did not find a statistically significant link between group size and prosocial behavior, they observed that the largest social group with the greatest number of calves was the most socially interconnected. Other recent studies have found more affiliative behaviors and fewer aggressive behaviors among related elephants (*Harvey et al., 2018*; *Finch et al., 2020*), supporting the recommendation for keeping elephants in related, multi-generational groups to optimize welfare. Using free-ranging elephants as a model for structuring elephant groups in zoos has several benefits, including promoting and facilitating social learning and natural social behaviors, reducing abnormal behaviors and stereotypies, and improving health, welfare, and reproduction (*Hartley, Wood & Yon, 2019*). However, to hold the greater number of elephants that comprise multi-generational groups, zoos would require far larger and more complex facilities. This presents a significant challenge due to the space and resources that would be required.

The social environment has a considerable impact on stereotypic behavior rates, an important indicator of compromised welfare (*Greco et al., 2016a*, *2017*). Spending more time with larger numbers of conspecifics and the amount of time spent with juveniles is associated with reduced stereotypy rates, whereas being housed separately increases stereotypic behavior (*Greco et al., 2016a*). Inter-zoo transfers also increase the risk because they can be disruptive to social groups and break important social bonds (*Clubb & Mason, 2002*; *Armstrong & Johnson, 2021*). Transfers are usually for breeding purposes, coordinated through programs such as the Species Survival Plan that AZA accredited zoos have adopted to promote genetic diversity (*Rees, 2011*). However, elephants can also be moved between facilities as a result of space limitations, sex of offspring (*Williams et al., 2019b*), or exhibit closure. The transfer of group members can affect social stability and limit long-term success in zoo social groups (*Williams et al., 2019b*). As many as 80% of the elephants in North American zoos in 2012 had experienced at least one inter-zoo transfer (*Prado-Oviedo et al., 2016*). *Clubb et al. (2008)* found that inter-zoo transfers lessened Asian elephant survivorship, an effect that lasted four years post-transfer, and Asian calves removed from their mothers at a young age tended to have poorer outcomes. Considering the complexity and importance of elephant social ties, it would be important to study the effects of transfers on elephants based on age, sex, and species.

Male elephants pose a particular challenge in captive facilities due to their strength, social needs, aggressiveness, and strong sexual and competitive motivations (*Lee & Moss, 2009*; *Hartley, Wood & Yon, 2019*). The proportional increase in the captive male population in recent years presents an additional challenge. Going forward, the number of males born will exceed the number that can be maintained in breeding groups and housing will be needed to accommodate multiple groups of males (*Readyhough et al., 2022*). In 2017, males made up only 21% of the Asian elephant population in North America but will eventually approach 50% (*LaDue et al., 2022b*), requiring greater consideration of their

physical and social needs and living space. *Schmidt & Kappelhof (2019)* suggest that Asian elephant husbandry guidelines in Europe may need to be redefined to, for example, encourage zoos to hold male offspring longer. Still, more space would be needed to hold males. They also suggest potentially requiring zoos to restrict breeding to keep the number of males manageable.

Increasing evidence indicates that male elephants display extensive social behaviors that rival the complexity of that of females (*LaDue et al., 2022b*). Despite their social nature, males are typically held separately from females and other males although some zoos have integrated adult males into social groups. In North America and Europe, the majority of males are kept at zoos with females but no other males, restricting social learning from older males and the development of appropriate social and reproductive behaviors (*Hartley, Wood & Yon, 2019*). Although the importance of sociality for male elephants is stressed by some (*LaDue et al., 2022b*; *Readyhough et al., 2022*), and although there are welfare benefits from social interactions (*e.g.*, reduction in stereotypies, *Readyhough et al., 2022*), progress appears to be slow. Some U.S. and European zoos have created all-male "bachelor" groups in response to the evidence for male sociality and the need to provide space for an increasing number of captive males (*Hartley, Wood & Yon, 2019*; *Readyhough et al., 2022*). In North America, there are currently two zoos holding bachelor groups (*AZA, 2019*, *2021a*); in Europe, there are nine (*Schmidt & Kappelhof, 2019*). However, *Hartley, Wood & Yon (2019)* caution that the focus of these groups should be on the social development of the elephants rather than on creating bachelor groups as a convenient solution to housing males. Subadult males of similar age and experience level who are held together in bachelor groups are at risk of developing abnormal behaviors or a limited behavioral repertoire due to the absence of social learning from mature males and family groups, including failure to learn reproductive behaviors (*Hartley, Wood & Yon, 2019*). Mixed-age bachelor groups that include a mature male may facilitate social learning (*Readyhough et al., 2022*). The social makeup of bachelor groups is subject to change, as males may be moved more often for breeding purposes (*Schmidt & Kappelhof, 2019*), potentially breaking social bonds. The effects of such moves on individual males and groups is an area for future study. Importantly, the creation of bachelor groups is a relatively new development in zoos; as such, it is too early to determine if it will be feasible over a long period of time (*Schmidt & Kappelhof, 2019*). Also, many zoos prefer to keep their male with a female group for his well-being (*Schmidt & Kappelhof, 2019*). Clearly, providing appropriate housing and social conditions for males will continue to be a challenge as the number of male elephants continues to increase.

Optimizing the social lives of captive elephants may involve a variety factors: increasing group size (*Lasky et al., 2020*), facilitating highly related multi-generational groups (*Hartley & Stanley, 2016*; *Harvey et al., 2018*; *Williams et al., 2019b*; *Finch et al., 2020*; *Armstrong & Johnson, 2021*), integrating males and females (*Hartley, Wood & Yon, 2019*; *Lasky et al., 2020*), developing fission-fusion housing (*Schmidt & Kappelhof, 2019*), and creating mixed-age bachelor groups (*Hartley, Wood & Yon, 2019*; *Readyhough et al., 2022*). These improvements all require significant resources, including larger, more complex

environments that allow for the management of males and females (*Hartley, Wood & Yon, 2019*). At the same time, changes must be carefully approached and science-based. For example, while fission-fusion housing strategies are suggested as a way to improve welfare, uncontrollable social changes and social frustration could generate locomotor stereotypies (*Greco et al., 2017*). More research is necessary to determine a management strategy that would enable fission-fusion through free choice of social associations and indoor/outdoor locations on a round-the-clock basis. Although the opportunity for appropriate social contact is considered to be more important for welfare than space (*Williams et al., 2020*), *Veasey (2020)* cautions that zoos with limited space will face challenges in addressing such important behaviors as elephant sociality, feeding, and foraging. Certainly, greater capacity is required to hold large or growing groups of elephants and to accommodate males (*Hartley, Wood & Yon, 2019*; *Schmidt & Kappelhof, 2019*). In addition, there is a need to better understand captive elephants' social networks and social behavior over longer periods of time and how social changes affect individuals and groups (*Williams et al., 2020*). The resource-intensive conditions (*e.g.*, space, finances, personnel) required to facilitate appropriate social conditions for captive elephants could stand in the way of optimizing welfare and could determine whether elephants should be kept in captivity.

## Feeding challenges

*Vicino & Miller (2015)* stress that animals should have the opportunity for a thoughtfully presented, well-balanced diet. For elephants, feeding involves much more than meeting dietary requirements. It often involves cognitive and physical challenges that are essential to an elephant's behavioral ecology and well-being. Foraging involves movement, exploration, and choosing areas in a landscape that hold high foraging opportunities, then homing in on preferred foods (*Das, Kshettry & Kumara, 2022*). The handling of food may involve manipulation (*e.g.*, prying bark from a tree, digging, kicking dirt off of grass roots) before consumption (*Poole & Granli, 2009*). Feeding is often done in concert with the family group, making it a social activity. Individual elephants vary their diets based on availability, preferences, and physiological needs (*Gill et al., 2023*). Free-ranging elephants are highly diverse feeders, consuming more than 100 seasonally and geographically varying food species (*e.g.*, grasses, trees, bark, roots, fruits, and aquatic plants; *Dierenfeld, 2006*; *Campos-Arceiz & Blake, 2011*; *Das, Kshettry & Kumara, 2022*). As large-bodied herbivores, elephants naturally spend 60–80% of their waking hours foraging over long distances (*Poole & Granli, 2009*). In contrast, zoo diets tend to be more restricted in terms of the variety of foods, are seasonally invariant, and (often excessively) high in calories (*Carneiro et al., 2015*; *Schiffmann et al., 2018*). Diets for elephants in zoos generally consist of dried forage, usually hay, supplemented with commercial concentrate feed pellets, vitamins, fruits, and vegetables (*Dierenfeld, 2006*). Feeding schedules tend to be temporally predictable with high spatial predictability and only moderate variation in feeding methods (*Greco et al., 2016b*), although some zoos try to diversify the way animals are fed as a form of limited enrichment (*e.g.*, puzzle/feeder balls, elevated nets with hay; *Ramírez et al., 2023*). Not unexpectedly, a more varied and unpredictable feeding regime

enhances well-being in captive elephants by promoting walking (*Holdgate et al., 2016*). Nearly three-quarters of the elephants across 65 AZA accredited North American zoos were determined to be overweight or obese, putting them at risk of health conditions such as ovarian acyclicity (*Morfeld et al., 2016*), joint disease/osteoarthritis (*West, 2006*), foot ailments, and gait problems (*Roocroft & Oosterhuis, 2001*).

Timing, frequency, and methods of food provision are key components of elephant feeding programs (*Greco et al., 2016b*). These areas nevertheless require continuous attention to avoid temporal and spatial predictability. As with enrichment, feeding programs require considerable keeper time, resources, and biologically informed policies. Including browse in the diet appears to be beneficial for elephant welfare as it increases opportunities for foraging behaviors and the amount of time spent feeding, decreases inactivity, and allows elephants to engage in naturalistic feeding behaviors (*e.g.*, manipulating branches, stripping leaves and bark; *Stoinski, Daniel & Maple, 2000*; *Lasky et al., 2020*), although it may not decrease stereotypic behaviors (*Lasky et al., 2020*). In highlighting the importance of foraging, AZA and BIAZA guidelines state that browse should be made available to elephants; however, neither organization specifies the amount of foraging time or type of foraging opportunities (*Veasey, 2020*). This is an area that must be addressed in order to standardize feeding practices and improve elephant welfare.

## CURRENT HEALTH AND WELFARE ISSUES

Although *Vicino & Miller*'s *(2015)* opportunity for optimal health would seem to be met with the presence of constant veterinary care, questions remain about whether the captive environment actually decreases the likelihood of elephants being healthy insofar as the welfare of elephants is multidimensional. A variety of variables may factor into the clinical problems elephants experience in the captive environment. In this section, we describe what is known about the current mental and physical health of captive elephants. We also discuss the impact that chronic stress has on mental and physical health, so that behavioral abnormalities and opportunistic infections become prevalent, sometimes leading to increased mortality. All these factors contribute to a generally problematic clinical situation for captive elephants.

### Brain and behavior
#### Stereotypies

A prevalent, observable abnormality (*Bacon, 2018*) in captive but not free-ranging animals is stereotypic behavior (*Mason & Latham, 2004*; *Mason & Rushen, 2008*). Stereotypies in elephants typically take several forms: whole body (*e.g.*, limb-swinging, back and forth movement), locomotor body (*e.g.*, pacing, repeatedly walking in the same pattern), self-directed (*e.g.*, limb or head banging, self-stimulating behaviors), and oral (*e.g.*, trunk sucking, bar biting) (*Glaeser et al., 2021*). Between ~47% and ~ 85% of elephants in zoos exhibit stereotypies, which can consume up to ~20% of the animal's daily activity (*Mason & Latham, 2004*; *Mason & Veasey, 2010*). *Greco et al. (2016a)* determined in North American zoos that, after feeding, stereotypic behavior was the second most commonly

performed behavior in elephants. Stereotypies are not just behavioral. They reflect an underlying dysregulation in the motor control systems of the brain, that is, a form of brain damage that leads to excessive, purposeless movements (*Jacobs et al., 2021*).

In an attempt to alleviate stereotypies, zoos often provide directed types of enrichment (*Ramírez et al., 2023*) which, unfortunately, remain insufficient as not all types of enrichment are equally effective (*Law & Kitchener, 2017*; *Lyn et al., 2020*). It has indeed been estimated that zoo enrichment is only effective in significantly reducing stereotypies 53% of the time (*Swaisgood & Shepherdson, 2005*). More recent research found that the frequency of stereotypic behavior decreased in Asian bull elephants when they were housed socially with other bulls rather than housed alone (*Readyhough et al., 2022*). It should nevertheless be noted that enrichment that is transient can actually create more stress and frustration for the animal (*Latham & Mason, 2010*). Zoo enrichment thus runs the risk of being an expedient attempt to treat the specific psychological/behavioral/neural problems that arise from the captive environment itself (*Jacobs et al., 2021*), and may fall well short of the broad-spectrum enrichment provided by the natural habitat (*Swaisgood & Shepherdson, 2005*; *Morgan & Tromborg, 2007*). With a brain that is among the largest on the planet, at ~5,000 grams (*Manger et al., 2009*; *Manger, Spocter & Patzke, 2013*), elephants are evolutionarily adapted to flourish in an extremely complex, stimulating environment, which traditional captive facilities cannot replicate. Several researchers have suggested that providing complex spaces and enhancing the social environment may reduce stereotypies (*Greco et al., 2017*; *Glaeser et al., 2021*; *Scott & LaDue, 2019*). Nevertheless, a more comprehensive, *a priori* solution would be to not put elephants in a captive environment that leads to stereotypic behavior.

## Physical health

### Nutrition and metabolism

Elephants ingest a large variety of foods in the wild to meet their nutritional needs (*Dierenfeld, 2006*; *Campos-Arceiz & Blake, 2011*; *Das, Kshettry & Kumara, 2022*), whereas captive diets are frequently monotonous and characterized by a lack of essential vitamins and minerals (especially vitamin E and iodine; *Dierenfeld, 2006*), a deficiency of fiber, and excessive calories (*Tsuchiya et al., 2023*). Combined with lack of exercise, the high caloric diet contributes substantially to widespread obesity among captive elephants (*Gupta, Sharma & Swarup, 2015*; *Brown et al., 2019*; *Tsuchiya et al., 2023*). Many zoos have added browse (including bamboo) to increase dietary fiber and as enrichment (*Tsuchiya et al., 2023*) as it can reduce obesity and colic (*Hatt & Clauss, 2006*). Of 240 elephants examined in North American zoos, 74% were overweight and 34% clinically obese (*Morfeld et al., 2016*). Obesity in elephants may be associated with poor health and reproduction (*Morfeld et al., 2014*). As in North American zoos, condition scores for the majority of elephants in European zoos indicated they were overweight (*Schiffmann et al., 2018*). Insofar as optimizing elephant nutritional intake for captive elephants can be problematic (*Schiffmann et al., 2018*), zoo diets remain a challenge.

### Skin and musculoskeletal health

One third of elephants with medical conditions in North American zoos have skin issues (*e.g.*, lesions, overgrowth of dead skin, parasites, sunburn, pressure sores; *Fowler, 2006a*; *Mikota, 2006*; *Brown et al., 2019*), although the prevalence of skin lesions in North American populations is reported to have decreased in the last 20 years with improved bathing and skin care protocols (*Edwards et al., 2019*). An elephant's skin requires daily care, which is achieved in the natural habitat by regular dust and mud bathing, abrasion, and occasional bathing (*Lehnhardt, 2006*). To this end, the AZA now requires that captive elephants have access to scratching posts and opportunities for bathing, dusting, and wallowing (*Edwards et al., 2019*; *AZA, 2021b*).

Foot disease is another common ailment of captive elephants. Early estimates suggested that ~50% of captive elephants suffer from foot ailments (*e.g.*, hyperkeratosis, cracked nails, infections), often from unsanitary or overly wet conditions (*Fowler, 2001*, *2006a*). However, one study across 13 U.K. zoos found that more than 80% of elephants demonstrated foot disease, with over 85% exhibiting gait abnormalities (*Harris, Sherwin & Harris, 2008*). More recently, *Miller, Hogan & Meehan (2016)* suggested that approximately two-thirds of elephants in the U.S. had foot issues. In a post-mortem imaging study, *Regnault et al. (2017)* found that every captive elephant (*n* = 21) they examined exhibited foot pathology. *Wendler et al. (2020)* state that rather than being the result of one or two factors, poor foot health may indicate a generally poorer husbandry system. They found that more advanced husbandry conditions (*e.g.*, large areas, high proportions of sand substrate) were associated with each other and better foot scores. More limited conditions (*e.g.*, more time spent indoors, higher proportion of hard substrate) were associated with each other but also indicated worse foot scores.

Deeper musculoskeletal ailments are also commonplace in captive elephant populations, including osteoarthritis, which regularly occurs prematurely in captive elephants. Such ailments are associated with pain and joint stiffness, inability to stand, and sometimes require euthanasia (*Issa & Griffin, 2012*; *Buckwalte et al., 2013*). *Miller, Hogan & Meehan (2016)* attributed musculoskeletal disease to time spent on hard substrates (*e.g.*, concrete, packed soil) and space experienced in indoor/outdoor exhibits ("space experience" is defined as the measure of space weighted by amount of time spent in that space). Although the authors found only 11.1% of the elephants in their study showed evidence of musculoskeletal problems, they state that the prevalence of those abnormalities was likely underestimated due to lack of sensitive diagnostic techniques.

### Dental disease

African elephants use their tusks for many purposes (*e.g.*, digging, carrying, sparring, debarking trees; *Weissengruber, Egerbacher & Forstenpointner, 2005*; *Dumonceaux, 2006*). Although both free-ranging and captive elephants are prone to tusk injuries (*Steenkamp et al., 2008*), these injuries are particularly common in captive elephants, who frequently encounter hard, unyielding materials (*e.g.*, concrete, metal). Tusk injuries have been documented in 31% of elephants across 60 North American zoos (*Steenkamp et al., 2008*).

Injuries involving the pulp cavity can become infected and lead to necrosis and fatality (*Dumonceaux, 2006*; *Rose et al., 2018*). Unlike carnivores, elephant molars are produced in the back of the jaw and grow forward continuously. Molar fragments must be shed at the front of the mouth with regularity to make room for the progressive movement. Access to an appropriate diet, including adequate roughage, contributes to healthy molar shedding (*Asquith-Barnes et al., 2017*). Unfortunately, molar tooth retention is a common dental problem for captive elephants (*Steenkamp, 2021*). This can cause pain and reluctance to eat, and frequently results in periodontal infection due to the impaction of food and bacteria around the tooth. Oral cavity infections can cause chronic and painful inflammation of the oral cavity (*i.e.*, stomatitis). Another functional abnormality that captive Asian elephants frequently suffer is malocclusion, which occurs when molar fragments rotate abnormally (*Steenkamp, 2021*). These functional abnormalities can contribute to complications (*e.g.*, uneven tooth wear, pain, disruption of mastication), including serious gastrointestinal disease and weight loss (*Dumonceaux, 2006*; *Steenkamp, 2021*).

### Digestive and gastrointestinal disease

It is now well understood that the gut microbiome of the elephant is impacted dramatically by stress and by changes in environment, in particular captivity (*Bo et al., 2023*). Numerous factors appear to be involved, including a lack of access to natural environmental microbes, dietary changes, frequent antibiotic use, and psychological/social stress (*Bo et al., 2023*). Unsurprisingly, gastrointestinal diseases are common in captive elephants (*Greene, Dierenfeld & Mikota, 2019*). Poor dietary management, coupled with lack of exercise, is a predisposing factor for colic (*Khadpekar et al., 2020*), especially in older elephants with decreased gut motility and dental problems (*Greene, Dierenfeld & Mikota, 2019*), which can progress to a life-threatening illness (*Khadpekar et al., 2020*). Other gastrointestinal diseases include diarrhea, constipation, bloat, gastritis, and occasionally intussusception or torsion. Infectious organisms are sometimes implicated in gastrointestinal conditions, including Salmonella, *Clostridium difficile*, and gastrointestinal parasites (*Greene, Dierenfeld & Mikota, 2019*).

### Infectious disease

Captive elephants are particularly susceptible to *Mycobacterium tuberculosis* (TB) and the elephant endotheliotropic herpesvirus (EEHV), which are highly contagious and can manifest as latent infections (*Mikota & Maslow, 2011*; *Fuery et al., 2018*; *Alkausar et al., 2024*). As with humans, TB is a deadly disease in elephants; however, unlike humans, elephants are often poorly tolerant of anti-tuberculosis drugs and treatment is often unsuccessful (*Lyashchenko et al., 2006*). TB was confirmed between 1994 and 2010 in 50 elephants in the U.S.—approximately 12% of the country's captive elephant population (*Mikota & Maslow, 2011*), with the actual number likely to be much higher because of subclinical carriers (*Greenwald et al., 2009*). Captive populations, particularly Asian elephants, are disproportionately affected by TB because of stress-induced immunosuppression (*Mikota, 2009*).

Although EEHV has been documented in free-ranging elephants (*Kerr et al., 2023*), it is particularly prevalent in captive and semi-captive conditions, especially in young, physically stressed or injured, or immunocompromised Asian elephants (*Schaftenaar et al., 2010*; *Alkausar et al., 2024*), including calves who have not received adequate maternal antibodies (*Hoornweg et al., 2021*). In this regard, group size appears to be an important variable insofar as larger groups increase the chance that calves will be exposed to EEHV-shedding elephants at a time when the calves still have high levels of maternal antibodies, in turn, protecting them from the virus (*Hoornweg et al., 2022*). This finding may be particularly important for the zoo environment, where captive group size remains low, reducing a calf's chances for exposure while still protected by maternal antibodies (*Hoornweg et al., 2022*).

Newly developed serological assays have shown that the prevalence of EEHV in captive elephants is nearly ubiquitous and, therefore, the risk of exposure is inevitable (*Hoornweg et al., 2021*). EEHV-HD (the haemorrhagic disease process associated with EEHV) is now the leading cause of death for captive Asian elephant calves under 8 years old (*Perrin et al., 2021*; *Alkausar et al., 2024*). It has killed over 100 captive and free-ranging elephants around the world in just over 30 years (*Long, Latimer & Hayward, 2015*), including 12–17% of all Asian elephant calves born in Western zoos (*Hoornweg et al., 2022*). The virus is presently responsible for 65% of all captive Asian elephant deaths in North America (*Titus et al., 2022*). In calves, the peak period of EEHV risk coincides with the weaning period (*Perrin et al., 2021*), with stressors contributing significantly to severe cases of EEHV that progress quickly and become fatal (*Kendall et al., 2016*). Specifically, the stress associated with social change triggered by the movement of animals between, into, or out of herds poses the greatest risk for recrudescent viral shedding, subsequent infection, and death by haemorrhagic disease in captive Asian elephant calves (*Titus et al., 2022*). Although EEHV has been mainly associated with Asian elephants, recent cases among captive African elephants have intensified concern about the susceptibility of this species to EEHV. Prior to 2019, only five cases of clinical disease from EEHV infection had been documented in African elephants. Since 2019, there have been at least seven EEHV cases in North American zoos, with three fatalities (*Pursell et al., 2021*).

In addition, infection caused by *Clostridium, Salmonella*, and *E. coli* species of bacteria are a frequent clinical challenge in captive elephants. Various Clostridium species are implicated in the deaths of elephants in zoos, including the organisms that cause botulism, as well as *C. perfringens*, *C. difficile*, and *C. septicum* (*Hess, 2022*). Salmonellosis tends to follow stress-related depression of the immune system (*Fowler, 2006b*), a finding that strongly suggests a negative relationship between captivity induced stress and immune competence (*Fowler, 2006c*; *Mikota, 2009*; *Schaftenaar et al., 2010*).

### Reproduction

Captive female elephants, because of stress and obesity, often enter prolonged periods of estrus acyclicity (*Hermes, Hildebrandt & Göritz, 2004*; *Edwards et al., 2015*). Other reproductive pathologies include a high incidence of ovarian cysts, and neoplasia (*Clubb & Mason, 2002*; *Hermes, Hildebrandt & Göritz, 2004*; *Brown, 2019*). Captive females tend to

reach sexual maturity at an accelerated rate, both in terms of estrus onset and first pregnancy (*Lee et al., 2016*). The accelerated rate of sexual maturity is often associated with early reproductive senescence (*Hermes, Hildebrandt & Göritz, 2004*). Although rare in free or semi-captive (*i.e.*, less intensive confinement and social restriction) populations, captive elephants suffer from a high rate of stillbirth, infant mortality, and infanticide, with a 20% stillbirth/perinatal death rate in North American zoos (*Taylor & Poole, 1998*) and 21% in European zoos (*Perrin et al., 2021*). *Saragusty et al. (2009)* found a skewed sex ratio (*i.e.*, more males are born in zoos), and that the juvenile mortality rate in captive facilities is almost double that in the wild. Indeed, captive Asian elephants are five to eight times more likely to deliver a stillborn calf than elephants managed in extensive systems (*e.g.*, logging camps) in Asia (*Clubb & Mason, 2002*), potentially because of obesity (*Taylor & Poole, 1998*). It has been suggested that modifications to captive environments and husbandry programs focused on enhancing social well-being, environmental complexity, nutrition, and exercise would help to ameliorate the widespread reproductive dysfunction observed in captive elephants (*Brown, 2019*).

### Longevity, survival, and mortality rates

The elephant population in zoos is generally not sustainable without imports from the wild (*Wiese, 2000*; *Hutchins & Keele, 2006*; *Mar, Lahdenperä & Lummaa, 2012*; *Kurt, 2014*) due to the poor reproductive success and low survivorship of captive elephants, who suffer from a reduced median lifespan as well as a higher mortality rate than their free-ranging and semi-captive counterparts (*Clubb et al., 2009*). *Schmidt & Kappelhof (2019)*, however, report that the Asian elephant EEP (European Association of Zoos and Aquaria Ex situ Programme) has become successful, with most demographic and genetic parameters showing healthy numbers of Asian elephants in the population. Although limited data exist on free-ranging Asian elephant lifespans, semi-captive Asian elephants used in the logging industry in Myanmar have a reported average longevity of between 34–41.7 years, with ~10% of females living to be over 60 years old (*Clubb et al., 2008*). Furthermore, reports from across Asian range countries document animals living well into their 80s (*Mobasheri & Buckley, 2021*). However, Asian and African elephants in zoos have a median lifespan of between 16.9–18.9 years (*Clubb et al., 2008*), with captive born Asian elephants having higher adult mortality rates than those born in the wild (*Clubb et al., 2008*). For African elephants, average life expectancy in the wild is between 41–56 years, with ~5% of individuals living to be over 65 years of age (*Clubb et al., 2009*; *Lee et al., 2016*). Captive African elephants exhibit a mortality rate that is 2.8 times higher than their free-ranging counterparts (*Clubb et al., 2008*). *Scherer et al. (2023)* examined historical data to determine changes in the survivorship of elephants in American and European zoos, as compared to findings by *Clubb et al. (2008)* that survivorship had improved for African elephants in zoos since 1960, but not for Asian elephants. *Scherer et al. (2023)* also determined that from 1960 to 2023, improvement in survivorship was significant for African elephants. They found there was close to a significant improvement for Asian elephants. Asian elephants generally had a higher survivorship than did African elephants.

Juvenile survivorship in zoos since 1960 did not change significantly; however, it was higher in African elephants likely due to the effects of EEHV on Asian elephants.

## A MORE GLOBAL PERSPECTIVE

Although the present review has focused primarily on Western facilities, the situation in other countries, where there is often less regulation, is considerably worse, with improvements even more urgently needed. For example, a visual assessment of 81 tourist and temple elephants in India between 2004–2005 found significant physical ailments, including 74% showing foot issues in the form of fissures, 20% of which were classified as severe, and 43% displaying hyperkeratosis (*Ramanathan & Mallapur, 2008*). *Vanitha, Thiyagesan & Baskaran (2010)* determined that temple elephants in Tamil Nadu, India, spend less than an hour a day walking and exercising and are chained nearly 70% of the time in a small space, raising concerns for foot disease, obesity, arthritis, and lowered life expectancy (cf., *Ramanathan & Mallapur, 2008*). Among captive elephants in temple systems in Tamil Nadu, India, the prevalence of stereotypies was documented at ~49% and was positively correlated with the duration of chaining (*Vanitha, Thiyagesan & Baskaran, 2016*). In Thailand tourist camps, 57% of elephants exhibited stereotypies, with the rates being highest in elephants between 4–10 years of age (*Fuktong et al., 2021*). Finally, although accrediting organizations (*e.g.*, AZA, BIAZA) strongly discourage general use of free contact, the bullhook (or ankus), and prolonged chaining, such management practices are still in use in facilities around the world. For example, *Bansiddhi et al. (2019)* reported that use of the ankus is a welfare concern among elephants used for tourism; half of the elephants studied had wounds, with most caused primarily by the ankus (*e.g.*, abrasions, lacerations, abscesses) but also knives (*e.g.*, penetrating and incision wounds), scratches, and pressure wounds. Captive elephants in Sri Lanka are also managed through subjugation, with chaining and liberal use of the ankus (*Fernando et al., 2011*). Unfortunately, government regulations are often inadequate to protect elephant welfare or have not been effectively implemented or enforced (*Fernando et al., 2011*; *Bansiddhi et al., 2018*; *Baker & Winkler, 2020*).

## ANIMAL WELFARE IMPLICATIONS

In the present review, we have highlighted concerns that, after a preliminary review of the literature, were determined to be important to elephant welfare in the captive setting. The present overview indicates that, for elephants, there is a considerable mismatch between the captive and natural environment, one that appears to negatively impact well-being in captivity (*Mason, 2010*; *Hosey, Melfi & Ward, 2020*). Moreover, the natural, sociobiological characteristics of elephants (*e.g.*, the need for space, cognitive and social complexity, and dietary practices) predict the welfare challenges summarized above (*Clubb & Mason, 2007*; *Miller, Andrews & Anderson, 2012*; *Pomerantz, Meir & Terkel, 2013*; *Hosey, Melfi & Ward, 2020*; *Mellor et al., 2021*), outcomes that were outlined in *Mason*'s *(2010)* comparative examination of species differences in response to captivity.

Whether zoos can provide for the needs of elephants is a question that has received increasing attention with regards to environmental enrichment (*Alligood et al., 2017*;

*Hosey, Melfi & Ward, 2020*; *Fernandez & Martin, 2021*; *Jacobs et al., 2021*; *Mellor et al., 2021*). The relationship between different forms of environmental enrichment and well-being is not well understood and, whereas some directed enrichment methods may appear to have a positive impact in specific ways (*Ramírez et al., 2023*), there is no evidence to suggest it allows species like elephants to flourish in captive facilities (*Meehan et al., 2019*). The effects of environmental enrichment can only be fully understood in the context of valid and reliable methods of welfare assessment (*Brereton & Rose, 2023*). To that end, there have been several recent advances in developing systematic evidence-based welfare assessment methods for captive elephants (*Meehan et al., 2019*; *Williams et al., 2019a, 2019b*; *Yon et al., 2019*; *Bansiddhi, Brown & Thitaram, 2020*; *Bansiddhi et al., 2020*). In a large-scale epidemiological study of 291 elephants across 70 North American zoos, the Elephant Welfare Initiative underscored the importance of larger social groups for elephant welfare (*Carlstead et al., 2013*; *Greco et al., 2016a*; *DiVincenti, McDowell & Herrelko, 2023*). Another large scale, meta-analysis of United Kingdom zoos found the best indicators for elephant welfare were reduced stereotypies, reduced glucocorticoids, and improved body condition scores, followed by increased lying rest and positive social interactions (*Williams et al., 2018*). Although these projects represent a step forward in captive animal management, it is important to note that assessing welfare is not equivalent to providing welfare or promoting well-being. It remains unclear just how extensively any of these proposed welfare measures are employed. Of those listed above specific to elephants, only *Yon et al. (2019)* actually states that the welfare measures have been implemented. Therefore, proposed improvements in assessment techniques are only the first step in addressing captive welfare issues and are not, by themselves, demonstrations of improved well-being (especially if they are never implemented).

In captivity, it is exceptionally difficult to meet elephants' overall ecological needs, as well as their individual needs (*Armstrong & Johnson, 2021*). Recently, some zoos have reported improvements in welfare, often related to exhibit expansion and changes in husbandry. These areas include increased movement (*Glaeser et al., 2021*), reduced stereotypies (*Finch et al., 2020*; *Glaeser et al., 2021*), improved feeding strategies/opportunities (*Glaeser et al., 2021*; *Lasky et al., 2020*) consideration of male social needs (*Readyhough et al., 2022*), increased activity and outdoor time (*Glaeser et al., 2021*), and end of bullhook use (*AZA, 2022*). However, for many zoos some of these improvements are aspirational (*e.g.*, bull groups-*Hartley, Wood & Yon, 2019*; *Finch et al., 2020*). Much of the literature we have cited acknowledges existing shortcomings and areas where further improvement is required. In fact, zoos continue to be limited by space and available human and financial resources (*Doyle, 2018*), which also limits the extent to which they can provide larger, more complex, and variable environments that allow elephants to engage in a full suite of natural behaviors (*e.g.*, fission-fusion), particularly as compared to elephant range countries. It is unclear yet how zoo improvements will affect health and well-being long-term, as certain problems persist (*e.g.*, stereotypies, lack of natural foraging opportunities, social group size and composition, male elephant housing, infectious diseases, foot pathologies, and obesity). As noted by *Pierce & Bekoff (2018)*, discussions of animal welfare in zoos tend to focus on incremental improvements without addressing the

underlying problems that captivity presents; they consequently call for completely changing the captive landscape.

From an ethical point of view, the deeper issue is whether elephants can ever thrive, not simply survive, in captivity. This entails understanding not only how physically healthy organisms are or how long they live, but how well individual animals live—the overall quality of their life, their well-being. It includes such dimensions as the ability to exercise autonomy (*Vicino & Miller*'s *(2015)* opportunity for choice and control) and be stimulated by significant challenges in the environment. Current evidence suggests that, although zoos can renovate existing enclosures with some physical and behavioral benefit (*Lucas & Stanyon, 2017*), they generally cannot provide a sufficient facsimile of a free-ranging life to allow captive elephants to thrive. Insofar as elephants cannot usually be released from a zoo into a free-ranging natural environment as they do not have the necessary survival skills, there are limited ethical options. *Atkinson & Lindsay (2022)* suggest that an enclosure size of 1 km$^2$ or more of environmentally complex natural habitat in a warm climate could provide captive elephants the chance for more fulfilling lives. *Poole & Granli (2009)* offer a scenario for facilities 50–75 km$^2$ in size that would hold two to three family groups (20 to 30 elephants to allow for the development of fission-fusion characteristics) and adult males to create a naturally functioning population. They point out, however, that the number of elephants would eventually outgrow the limitations of such a facility, necessitating ethically unacceptable interventions such as transfers of individuals to other facilities.

On a global scale, some countries are reevaluating whether they should keep elephants in captivity or provide additional protections for captive elephants. In 2021, for example, the United Kingdom began ongoing discussions on banning elephants in zoos because of the challenges of providing them with a healthy environment (*Aoraha, 2021*; *Atkinson & Lindsay, 2022*). Proposed legislation in Canada (*i.e.*, Bill S-15, introduced in 2023) aims to prohibit possessing, breeding, or impregnating elephants (and great apes) in captivity and to prohibit using them for entertainment purposes (*Senate of Canada, 2023*). The ongoing challenge of housing elephants in zoos is reflected in the fact that, according to AZA regional studbooks for Asian and African elephants, the number of AZA accredited zoos holding elephants in the U.S. appears to have dropped from 67 to 49 in the last decade. Indeed, since 1991, 34 AZA accredited North American zoos have ended their elephant exhibits (Table 1). They have done so for a variety of reasons (*e.g.*, lack of space, funding, lack of social opportunities, inability to provide for aging elephants, non-compliance with AZA social group size standards). These cessations have resulted in the transfer of 52 elephants, with 31 initially transferred to other zoos (with two animals subsequently being moved to a sanctuary) and 23 transferred directly to sanctuaries. If an elephant is healthy enough to transfer, accredited sanctuaries represent a viable alternative. Although still a form of captivity, authentic sanctuaries provide a permanent home and a larger, more complex natural environment. In this regard, it was recently reported that stereotypic behaviors in several captive African elephants reintegrated into the wild immediately ceased upon their release (*Pretorius, Eggeling & Ganswindt, 2023*), but more research is clearly needed on the welfare effects of moving elephants to a sanctuary setting. Accredited

**Table 1 AZA accredited zoological facilities in North America that have terminated their elephant exhibits since 1991.**

| Zoological facility[1] | Final year | Elephant[2] | Transfer to[3] | Status[4] |
|---|---|---|---|---|
| Sacramento Zoo (CA) | 1991 | F Asian (Winky) | Detroit Zoo, then PAWS in 2005 | d. 2017 |
| Louisiana Purchase Gardens and Zoo (LA) | 1999 | F Asian (Shirley) | TES | d. 2021 |
| Mesker Park Zoo (IN) | 1999 | F Asian (Bunny) | TES | d. 2009 |
| Henry Vilas Zoo (WI) | 2000 | F Asian (Winkie) | TES | d. 2017 |
| Greater Vancouver Zoo (CAN) | 2003 | F Asian (Tina) | TES | d. 2004 |
| Chehaw Wild Animal Park (GA) | 2004 | F African (Tange) | TES | alive |
| | | F African (Zula) | TES | d. 2009 |
| San Francisco Zoo (CA) | 2005 | F Asian (Tinkerbelle) | PAWS in 2004 | d. 2005 |
| | | F African (Lulu) | PAWS | d. 2024 |
| Detroit Zoo (MI) | 2005 | F Asian (Winky) | PAWS | d. 2008 |
| | | F Asian (Wanda) | PAWS | d. 2015 |
| Lincoln Park Zoo (IL) | 2005 | F African (Wankie) | Hogle Zoo | d. euthanized shortly after arrival |
| Gladys Porter Zoo (TX) | 2006 | F African (Ruth) | Milwaukee Zoo | alive |
| Abilene Zoo (TX) | 2007 | F African (Tanya) | Cameron Park Zoo | d. 2020 |
| Philadelphia Zoo (PA) | 2009 | F African (Kallie) | Pittsburgh Zoo's Int'l Conservation Center, then Cleveland Zoo in 2011 | alive |
| | | F African (Bette) | Pittsburgh Zoo's Int'l Conservation Center | alive |
| | | F Asian (Dulary) | TES | d. 2013 |
| Brookfield Zoo (IL) | 2010 | F African (Joyce) | Six Flags Wild Safari Adventure, NJ | alive |
| Lion Country Safari (FL) | 2010 | F African (Ladybird) | Greenville Zoo | d. 2014 |
| | | M African (Bulwagi) | Disney's Animal Kingdom in 2006, then Birmingham Zoo in 2010 | alive |
| | | F African (Mama) | Dallas Zoo | d. 2015 |
| | | F African (Stumpy) | Dallas Zoo | d. 2012 |
| Jackson Zoo (MS) | 2010 | F African (Juno) | Nashville Zoo, then TNEC in 2015 | d. 2015 |
| | | F African (Rosie) | Nashville Zoo, then TES in 2015 | d. 2016 |
| Central Florida Zoo (FL) | 2011 | F Asian (Maude) | Zoo Miami | d. 2013 |
| Toronto Zoo (CAN) | 2013 | F African (Iringa) | PAWS | d. 2015 |
| | | F African (Thika) | PAWS | alive |
| | | F African (Toka) | PAWS | alive |
| BREC's Baton Rouge Zoo (LA) | 2013 | F Asian (Bozie) | Smithsonian's National Zoo | alive |
| Calgary Zoo (CAN) | 2014 | M Asian (Spike) | Busch Gardens Tampa Bay in 2013, then Smithsonian's National Zoo in 2018 | alive |
| | | F Asian (Kamala) | Smithsonian's National Zoo | alive |
| | | F Asian (Swarna) | Smithsonian's National Zoo | alive |
| | | F Asian (Maharani) | Smithsonian's National Zoo | alive |
| Greenville Zoo (SC) | 2014 | F African (Joy) | Cheyenne Mountain Park Zoo | d. en route to zoo |

(Continued)

| Zoological facility[1] | Final year | Elephant[2] | Transfer to[3] | Status[4] |
|---|---|---|---|---|
| Woodland Park Zoo (WA) | 2015 | F Asian (Chai) | Oklahoma City Zoo | d. 2016 |
| | | F Asian (Bamboo) | Oklahoma City Zoo | d. 2022 |
| Lee Richardson Zoo (KS) | 2015 | F African (Missy) | Cheyenne Mountain Park Zoo | alive |
| | | F African (Kimba) | Cheyenne Mountain Park Zoo | alive |
| Nashville Zoo (TN) | 2015 | F African (Sukari) | TES | alive |
| | | F African (Hadari) | TES | d. 2017 |
| | | F African (Rosie) | TES | d. 2016 |
| Virginia Zoo (VA) | 2016 | F African (Cita) | Zoo Miami | d. 2019 |
| | | F African (Lisa) | Zoo Miami | d. 2017 |
| Buffalo Zoo (NY) | 2018 | F Asian (Surapa) | Audubon Zoo | alive |
| | | F Asian (Jothi) | Audubon Zoo | alive |
| Riverbanks Zoo (SC) | 2019 | F African (Belle) | Milwaukee Zoo | alive |
| Santa Barbara Zoo (CA) | 2019 | No surviving elephants | | |
| San Antonio Zoo (TX) | 2023 | F Asian (Nichole) | TES | d. 2023 |
| Hogle Zoo (UT) | 2023 | F African (Christie) | Kansas City Zoo | alive |
| | | F African (Zuri) | Kansas City Zoo | alive |
| Knoxville Zoo (TN) | 2023 | F African (Edie) | TES | alive |
| | | F African (Jana) | TES | d. 2023 |
| El Paso Zoo (TX) | 2024 | No surviving elephants | | |
| Oakland Zoo (CA) | 2024 (announced July 2024) | M African (Osh) | TES | alive |
| Point Defiance Zoo (WA) | 2024 | No surviving elephants | | |
| Louisville Zoo (KY) | 2025 (announced March 2024) | F Asian (Punch) | TES | alive |
| | | F African (Mikki) | TES | alive |

**Notes:**

[1] Not included here are the Niabi Zoo (IL) and Six Flags Discovery Kingdom (CA), which lost/relinquished AZA accreditation prior to terminating their elephant exhibits.

[2] F, female; M, male.

[3] TES, The Elephant Sanctuary in Tennessee; PAWS, Performing Animal Welfare Society (CA); TNEC, The National Elephant Center (FL).

[4] d = year of death; alive as of 7/6/2024.

sanctuaries offer a legitimate option for elephants in need of rescue or placement, as evidenced by the number of elephants sent from North American zoos to these facilities. Currently, the number of elephant sanctuaries is limited, with two accredited facilities in the U.S. (*Global Federation of Animal Sanctuaries, 2024*) and few others around the world.

## CONCLUSION

In conclusion, we have focused on research conducted at accredited zoos, where higher levels of welfare are expected and indeed mandated. A review of the literature suggests that accredited zoos today are more focused on welfare and make an effort to enrich the lives of their elephants. Nevertheless, the evidence demonstrates that serious health and welfare

challenges persist, suggesting that, for practical and ethical purposes, elephants are generally unsuited to captive conditions. The declining number of AZA accredited zoos housing elephants speaks to the many challenges of keeping them in captivity. As we move forward with increased interest in the overall well-being of these and other animals, it will be important to recognize when certain environments cannot provide what some species need to flourish, and to consider science-informed policies determining which species could continue to be bred and kept in zoos and other captive facilities, and which species should not.

### Funding
The authors received no funding for this work.

### Competing Interests
Catherine Doyle is the director of science, research, and public policy at the Performing Animal Welfare Society (PAWS). Heather Rally is the founder of and veterinarian at Thrive Wild. Lester O'Brien is the founder of Palladium Elephant Consulting Inc. Lori Marino is an adjunct professor at New York University, and is president of the Whale Sanctuary Project.

### Author Contributions
- Catherine Doyle conceived and designed the research, analyzed the data, prepared figures and/or tables, authored or reviewed drafts of the article, and approved the final draft.
- Heather Rally performed the research, analyzed the data, authored or reviewed drafts of the article, and approved the final draft.
- Lester O'Brien performed the research, analyzed the data, authored or reviewed drafts of the article, and approved the final draft.
- Mackenzie Tennison analyzed the data, authored or reviewed drafts of the article, and approved the final draft.
- Lori Marino conceived and designed the research, analyzed the data, authored or reviewed drafts of the article, and approved the final draft.
- Bob Jacobs conceived and designed the research, analyzed the data, prepared figures and/or tables, authored or reviewed drafts of the article, and approved the final draft.

### Data Availability
This is a literature review.

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
