# Peer review of "Continuing challenges of elephant captivity: the captive environment, health issues, and welfare implications"

_PeerJ, doi:10.7717/peerj.18161_

## Round 0.1 · original submission · Major Revisions

Thank you for submitting this interesting review to PeerJ. I regret that I am unable to accept the manuscript for publication, at least in its present form. However, I am prepared to consider a substantially revised version that very carefully addresses the concerns highlighted by the reviewers. All questions and comments must be addressed comprehensively in a new version. Such a revised manuscript is likely to be reviewed again and there is no guarantee of acceptance. When you revise the manuscript, please prepare a detailed explanation about how you have dealt with all the reviewer comments, as well as my own ones.

"(with a focus on Western facilities)". This part of the title could be removed and explained (why focus mainly on the West) instead in other parts of the manuscript (e.g. Abstract, Introduction, Methods, Discussion). Please expand the remaining title text so that it more accurately reflects the contents of the review.
L41. Please provide further details (here or elsewhere) about the numbers and proportions of elephants kept at different types of sites.
L41-43. What about social factors (other elephants)? e.g. Living with family vs non-kin.
L63. What proportions of these 17,000 are held in the Global North vs South?
L92. Many readers may associate the term "survey" with a questionnaire-type study, which was not done.
L113. "a large quantity". Please be specific.
Line 115. "adequate space". Please be specific.
L158-159. "could provide". Again, this is too vague. Please explain clearly.
L205-207. It would be better to start the paragraph with a version of this text.

Reviewer 1 ·

Basic reporting

This review of elephant captivity and welfare is novel, interesting and reaches across disciplines, and I applaud the authors’ attempt to undertake such a review. This topic has not been reviewed for several years, and is therefore useful and important, and the review is very well written, with clear English.

I do feel the Introduction could benefit from some additional work, however, to provide greater background and context, and to set out the aims of the paper more clearly.

The introduction section felt a bit muddled in its structure. To make the argument easier to follow, I suggest reorganising this section to first frame the broad problem (e.g.: whilst animal welfare is mandated in many countries, many zoos or captive animal facilities are unregulated, and even those that are regulated may still struggle to look after large mammals adequately), and only then move on the specifics of the situation for elephants in captivity, finishing with a clear statement of the aims (and research question?) of the review.

Experimental design

I believe this review is within the aims and scope of the journal, and the review is generally well-organised, but I think more work is required in the Methods section (section 2) to fully explain the investigation and aim of the review.

Specifically, it would be useful to see more information about the aim for and use of the literature reviewed in Section 2. I recognise that you say that no filtering or selection occurred, but does that really mean that you reviewed and included information from all literature published between 2000 and 2024, that fit with the keywords listed?

What information were you extracting from all those papers? Was it to answer a specific research question? Were the parameters you use to organise sections 3-6 derived from the literature review, or did you already have those headings in place before reviewing the literature (and if so, how were those headings generated, and does that mean you did select literature that conformed to those headings)? It is unclear, currently, what information you already had, and what information/points has been generated specifically by the literature review.

Some of this detail can be provided in the Introduction section, and some would fit better in this Methods section. But either way, clearly setting out all of these details is important, so readers know if this was a systematic review, or a scoping review, or a simpler essay, for example. I think this is very important, as it will help to eliminate or pre-empt any claims of bias that would severely detract from the potential impact of the paper.

Validity of the findings

I fully agree with the authors that the welfare of captive elephants is compromised, and I know there is substantial evidence supporting this. But given my concerns about how the review was conducted (or rather, how it was reported), it is difficult to avoid feeling that there may be some bias in how the literature review is presented (which is what I hope my comments in section 2 above will reduce for readers of PeerJ).

That being said, the authors do a good job of highlighting unresolved questions and gaps in knowledge.

Additional comments

Minor points.

Lines 51 and 58-59 seem a bit contradictory, perhaps the situation could be clarified?

Line 67: 280 elephants in how many zoos, and how many of those are AZA accredited?

Lines 158-159: It would be useful to explain how/why Atkinson and Lindsay got to the figure of 100ha for enclosure sizes (I don’t disagree with them, but for readers unfamiliar with this topic, I think such a large number needs to be explained).

Line 210-212: I agree that sensory experiences are limited for elephants in zoos, given the restricted environment. But in the literature reviewed, was there any mention of how visitors/tourists may contribute (positively or negatively) to the sensory environment of the elephants? I know some research looked at differences in behaviour of zoo animals during the covid pandemic, for example, when no visitors were present: might work like this give us any clues as to whether or not zoo visitors can be stimulating for elephants?

Line 291-294 (and other mentions of stereotypies): I can’t remember if the Greco paper cited here, or other work which discusses stereotypies, described exactly which stereotypies are seen, and which ones, reduce with improved social conditions. If so, I think it would be interesting to report that detail here.

Lines 295-304: Is there any evidence that females (for example) do worse following inter-zoo transfers than males (who naturally disperse from their natal family)? Or any evidence to suggest an age impact – for example, if zoos move males before adolescence (when they typically show independence from their family), does that have more negative consequences than when moving when older? I appreciate that there may not be much published data on this, but raising these points as questions that need further investigation may be useful, to demonstrate to readers the complexity involved, as is done in the subsequent, related paragraphs (beginning on lines 306 and 319).

Section 4.1.1: Including Greco’s suggestion here improving the social environment and spatial complexity can reduce locomotor stereotypies is interesting, but I wonder if there is any evidence behind this suggestion that could also be mentioned/reviewed? For example, is there evidence that elephants housed in larger social groups or larger spaces (such as at Dublin or San Diego) exhibit fewer stereotypies than elephants in very small or solitary groups?

Reviewer 2 ·

Basic reporting

The manuscript provides a summary of the past 24 years of publications on elephant welfare under human care and its associated challenges. The authors looked into different conditions of captive care that may impact welfare, emphasizing the lack of resources to provide appropriate habitat for elephants in zoos and ultimately questioning the future of these animals’ presence in zoological facilities.

The manuscript is presented in a clear structure that summarises a great number of works of literature. The authors summarise the literature on different conditions of captive care. In the introduction, the authors mention five main points for evaluating captive welfare. It would be better to structure the manuscript following these points instead of starting with space and walking distance. Later in this section, the authors also acknowledge that space alone may not be a good indicator. Alternatively, if the authors prefer to keep the current structure, it would be helpful to state at the beginning of each section which of the five main points the discussed section falls under.
Modern welfare practices focus on mental and physical welfare and the connection between the two. The manuscript provides a detailed summary of the physical aspect of welfare.
While the challenges of captive care are not new to science, modern zoological facilities have made progress towards improvement. As the authors also acknowledge that many of these animals will remain under human care for the rest of their lives, it would be beneficial if the authors summarised the potential solutions at the end of each section. This would also improve the impact and novelty of their findings.

Experimental design

It would be informative to present the total number of papers used so the percentage values would be more meaningful. I would also suggest including a table that lists the number of papers for each keyword/topic so that the readers would have a better sense of the materials included.

The authors list sanctuaries in the list of keywords. While there is definitely an overlap between the two, some zoos serve as sanctuaries. Nonetheless, the two types of facilities have different purposes, and as such, they should not be evaluated as one. Sanctuaries provide a home for old/injured animals that cannot be rehomed/released. Therefore, the animals in these facilities may be in worse shape, which could be due to their pre-existing conditions.

Validity of the findings

Overall, a good summary paper, and its conclusions are definitely important when it comes to the future of elephants under human care. Despite the limitations and access to resources, it is the duty of the care facilities to provide the best possible care for their animals (life-long for those that cannot be rehoused). For that reason, it would be helpful not only to present the issues and challenges but also to emphasise practices that can improve the welfare of these animals.

Annotated reviews are not available for download in order to protect the identity of reviewers who chose to remain anonymous.

---

## Round 0.2 · Minor Revisions

Thank you for carrying out detailed revisions to the manuscript. The revised version was seen by one of the original reviewers, and a new one. Please bear in mind that, overall, it has been quite challenging to find reviewers for this manuscript (despite the high profile of captive elephant welfare), and the summer holidays period was probably an additional factor. The reviewers have provided useful advice for some further minor but important revisions to the manuscript. I would be very grateful if these could be carried out.

Reviewer 2 ·

Basic reporting

The authors made a considerable effort during their review, and most of my comments were addressed. They extended the referenced literature and included additional references on cognitive enrichments. Furthermore, they included recommendations for better practices.

Experimental design

L 103-109: Actually, Martelli & Krishnasamy, 2023, states the exact opposite of what the authors state in this paragraph. They emphasise the importance of differentiating between the two terms and not to use it interchangeably:

“We suggest that by narrowing the definition of animal welfare rather than by expanding it, and by differentiating animal welfare from animal wellbeing and rejecting the equivalence of these terms, "

"In the text hereafter, animal wellbeing refers to the current physical, mental and social health of an individual or group of animals. Wellbeing is a state internal to the animal and is not equivalent to animal welfare. The term animal welfare refers to the systems of management employed to achieve the maximum potential wellbeing of the individual or group of animals in zoological institutions."

The methodology is much improved with a more detailed description on how the authors reviewed the literature and generated the parameters for the paper.

Validity of the findings

no comment

Additional comments

Please correct typos, double spaces, and ensure consistency in spelling, for example, problem-solving versus problem solving.

·

Basic reporting

Clear, unambiguous, professional

Experimental design

Rigorous and sound presentation of digested literature.

Validity of the findings

Arguments well developed. Conclusions sound.

Additional comments

The authors have already revised this paper extensively. I will therefore attempt to address only those areas of significant concern that might remain. Most are suggestions for minor tweaks to emphasize the relevance of this study.
Basic reporting is competent and well-articulated.
Design seems clear and sensible.
Validity of the findings: these contribute to the literature showing that a rethink of elephant keeping is necessary!

Specific comments to the authors:
57: The UK does have legislative oversight over specific zoo species: Secretary of State’s Standards for Modern Zoo Practice, which now includes a section on elephants. Could this not be rephrased to:
National and international Zoo associations have power over accreditation, but not necessarily over licensing. Licensing requirements for best practice addressing elephant welfare are currently limited (but see the UK’s Secretary of State’s Standards for Modern Zoo Practice).
62: choice and challenge (since in this reviewer’s opinion appropriately challenging environments are missing in all captive contexts).
70: does Africa here include the Middle East? Arab states are currently consumers of wild elephants for zoos.
A general question: has elephants.se updated its data recently?
88: do not fare well (delete “appear”). They die. It’s more than apparent.
108: still not clear what has changed to make this paper necessary. Has welfare improved (not really), are fewer elephants in captivity (still many breeding programs), is the negative publicity about captivity decreasing (not). Relevant to exclusion of papers prior to 2000 (line 147). Has anything changed for elephants?
Maybe: Considering these new perspectives on welfare and the continuing challenges for captive elephants…
115: is the UK not in the global north? Maybe: we exclude Mexico and the UK because…
129: play foraging = play, foraging?
175: why would we actually want large groups of elephants in captivity? (you discuss the issues of outgrowing space later) Larger is not better: groups that sustain positive and challenging social interactions whether large or small need space. My concern here is that “forcing” better welfare via large groups equals more captive animals.
216: stall size. The term alone says it all. Free choice associations and locations indoor and out on a 24/7 basis is required, not stalls.
228: Atkinson & Lindsay (2022) conclude that range countries are the only places where elephants can truly flourish. Move sentence to end of pp and delete However. The sentence summarises the problem.
257: in exhibit design changes or by going out of keeping elephants (please at least mention this as an option earlier – it’s in the final section).
391: to sustain high elephant welfare (while it is recognised that welfare is compromised by most researchers, zoos are not necessarily in agreement that “their elephants’” welfare needs improving?)
454: you are no doubt aware that interactive management (free contact) is being totally phased out in the EAZA context, since it represents welfare challenges to elephants and keepers….
469: female elephants typically remain with their natal herd, forming strong lifelong bonds with related females although family fissions are also common.
492: of course, if the calves die (as many do) the group size increases are temporary. There has been no recent longitudinal demographic analysis of elephant zoo groups to my knowledge.
579: see comment above about 24/7 unconstrained choice.
613: ovarian acyclicity, joint pain, foot and gait problems…
670: stereotypies are learned behaviours; even with the best environments, an elephant who has experienced 20-40 years of captivity and learned to stereotype may never “unlearn” this behaviour. My point is that one should be wary of arguing that extinguishing stereotypies is possible with environmental changes and represents welfare. The key necessary change is to provide environments so that the individuals never start!
683-4: note the repetition with discussion of obesity in pps above.
4.2.2: please note that there is no mention of condition. Muscle tone is often poor (hence the unable to get up issues).
954: The UK started the Elephant Welfare Group in response to Clubb & Mason, in 2010. The acceptance of recommendations is ongoing. So much time, so little change.
981: expected and indeed mandated.

---

## Round 0.3 · accepted · Accept

Thank you for carrying our the latest revisions. I am happy to accept this version, subject to some final edits. Please remove the two web links inserted into the Introduction (paragraph one), and Lines 468/469, and instead create appropriate references in the references list.